# CircADAMTS16 Inhibits Differentiation and Promotes Proliferation of Bovine Adipocytes by Targeting miR-10167-3p

**DOI:** 10.3390/cells12081175

**Published:** 2023-04-17

**Authors:** Chunli Hu, Xue Feng, Yanfen Ma, Dawei Wei, Lingkai Zhang, Shuzhe Wang, Yun Ma

**Affiliations:** Key Laboratory of Ruminant Molecular and Cellular Breeding of Ningxia Hui Autonomous Region, College of Animal Science and Technology, Ningxia University, Yinchuan 750021, China

**Keywords:** adipocyte proliferation, bovine, circADAMTS16, differentiation, miR-10167-3p

## Abstract

Circular RNAs (CircRNAs) are covalently closed-loop non-coding RNA (ncRNA) molecules present in eukaryotes. Numerous studies have demonstrated that circRNAs are important regulators of bovine fat deposition, but their precise mechanisms remain unclear. Previous transcriptome sequencing studies have indicated that circADAMTS16, a circRNA derived from the a disintegrin-like metalloproteinases with the thrombospondin motif 16 (*ADAMTS16*) gene, is high expressed in bovine adipose tissue. This gives a hint that the circRNA may be involved in the process of bovine lipid metabolism. In this study, the targeting relationship between circADAMTS16 and miR-10167-3p was verified using a dual-luciferase reporter assay. Then, the functions of circADAMTS16 and miR-10167-3p in bovine adipocytes were explored through gain-of-function and lose-of-function. The mRNA expression levels of genes were detected by real-time quantitative PCR (qPCR), and lipid droplet formation was phenotypically evaluated by Oil Red O staining. Cell proliferation and apoptosis were detected using CCK-8, EdU, and flow cytometry. We demonstrated that circADAMTS16 targeted binding to miR-10167-3p. The up-regulation of circADAMTS16 inhibited the differentiation of bovine preadipocytes, and the overexpression of miR-10167-3p promoted the differentiation of bovine preadipocytes. Meanwhile, CCK-8 and EdU results indicated that circADAMTS16 promoted adipocyte proliferation. Subsequently, flow cytometry analysis showed that circADAMTS16 promoted cell transition from G0/G1 phase to S phase, and inhibited cell apoptosis. However, up-regulation of miR-10167-3p inhibited cell proliferation and promoted apoptosis. Briefly, circADAMTS16 inhibited the differentiation and promotes the proliferation of bovine adipocytes by targeting miR-10167-3p during bovine fat deposition, which provides new insights into the mechanism of circRNAs regulation of beef quality.

## 1. Introduction

Among many important factors affecting meat quality, the content of intramuscular fat (IMF), or marbling, is essential to improve the flavor and palatability of beef. Meanwhile, the IMF content is also an important index of meat quality [1]. There are two different primary mechanisms for the expansion of adipose tissue, namely increasing the number of adipocytes (proliferation) and/or increasing the volume of adipocytes [2,3]. Fat formation is generally divided into two stages. The first stage is the differentiation of embryonic stem cells into mesenchymal stem cells with multiple differentiation potentials and the second stage is the terminal differentiation stage. Preadipocytes are characterized by mature adipocytes and can respond to hormones such as lipid droplets and insulin [4]. Fat deposition involves many transcription factors, enzymes, hormones, and signal pathways, and is a complex physiological process. Among them, peroxisome-proliferator-activated receptor γ (PPARγ) and CCAAT-enhancer binding protein α (C/EBPα) are two of most important regulatory factors in the development of adipogenesis. PPARγ actively regulates C/EBPα and promotes preadipocyte differentiation [5,6]. Proliferation is a short process that is necessary for the formation of preadipocytes, which are crucial for preadipocytes maturation [7]. Recent studies have shown that ncRNAs, including microRNAs (miRNAs), long non-coding RNAs (lncRNAs) and circRNAs, are capable of participating in the process of lipid metabolism, including the proliferation and differentiation of adipocytes, the formation of fat droplets, and the activation of adipocyte-specific genes, as well as the regulation of genes during the development and remodeling of adipose tissue [8,9,10]. In goats, miR-10a-5p targets Krüppel-like factor 8 (*KLF8*) and inhibits differentiation of intramuscular preadipocytes [11]. Through attenuation of phosphorylation of mitogen-activated protein kinase (MAPK)-p38 and MAPK-extracellular signal-regulated kinase 1/2 (ERK1/2), lnc-FR332443 positively regulates *Runx1* expression and suppresses adipocyte differentiation in mouse adipocytes [12]. Additionally, circ-PLXNA1 inhibits the differentiation in duck adipocytes [13].

CircRNAs are endogenous covalently closed-loop biomolecules present in eukaryotes, and belonging to the group of non-coding RNA molecules. In 1976, circRNA was first discovered in a viroid [14]. CircRNA is produced using a reverse spliceosome mechanism, regulated by cis- and trans-acting factors [15,16]. The circRNA has both tissue-specific and cell-specific expression patterns that play important biological functions in regulating protein function or byself-translation [17]. CircRNAs regulate gene expression by affecting transcription, mRNA renewal, and translation [18]. Notably, circRNA can participate in the splicing of target genes and translate the genes into peptides. Studies have shown that circRNA with an infinite open reading frame (ORF) can be amplified in a rolling cycle in an internal ribosome entry site (IRES)-independent manner, resulting in a production efficiency nearly 100 times higher than that of a linear transcript [19]. There are a variety of circRNAs and their action mode is relatively complex, and closely related to the distribution of circRNA in cells. It has been found that the circRNA located in the cytoplasm can serve as a molecular sponge to adsorb miRNA and regulate the mRNA transcription level, a function widely studied in the regulation of lipid metabolism [20]. CircRNA binds miRNAs through a sponge, preventing miRNAs from bindings to their target genes to regulate mRNA expression by blocking their binding [21]. For instance, circSAMD4A promotes the differentiation of preadipocytes by acting as an miR-138-5p sponge in humans [22], while the sus_circPAPPA2 inhibited the fat deposition of ovariectomized pigs through the miR-2366/glycoprotein k (GK) pathway [23,24]. The circRNA/miRNA interaction reveals a new epigenetic regulatory layer. The regulatory network composed of circRNA–miRNA–mRNA plays an important role in the regulation of lipid metabolism.

The disintegrin-like metalloproteinases with the thrombospondin motif (ADAMTS) is a large family of metalloproteases that participate in many physiological processes. Numerous pathologies are associated with abnormally expressed or imbalanced *ADAMTSs*, including cancer, inflammation, neurodegeneration, and cardiovascular disease [25]. The high expression of *ADAMTS15* is associated with a decreased proliferation and migration of cancer cells [26]. *ADAMTS1* inhibits angiogenesis in lung cancer cells through the PI3K/Akt-eNOS-VEGF pathway [27]. Previous studies have shown that knockdown of the *ADAMTS16* gene inhibits the proliferation of esophageal squamous cell carcinoma cells [28]. However, the regulation of bovine adipocyte deposition by the ADAMTS family has not yet been reported. circADAMTS16 is derived from the *ADAMTS16* coding region, and has a total length of 550 bp. Transcriptome sequencing results showed that circADAMTS16 was highly expressed in bovine adipose tissues (the complete sequence is shown in Appendix A), indicating that it might regulate the differentiation and fat-formation of bovine adipocytes. We amplified the interface, and the first generation sequencing results were consistent with the sequences obtained by sequencing, so the circADMTS16 obtained by sequencing was a real circular RNA [29]. The purpose of this study was to investigate the effects of circADAMTS16 on bovine fat deposition and its molecular regulation, so as to provide a basis for further research on bovine fat deposition.

## 2. Materials and Methods

### 2.1. Ethic Statement

Animal experiments were conducted according to the Regulations for the Administration of Affairs Concerning Experimental Animals (Ministry of Science and Technology, China, 2004). All animal protocols were approved by the Animal Ethics Committee of Ningxia University (permit number NXUC20200618) and Zerui Ecological Breeding Farm (permit number ZR20200615) [30].

### 2.2. Animal Samples

The adipose tissues of the primary adipocytes isolated in this experiment were from Zerui Ecological Breeding Farm (Yinchuan, China). Fresh back subcutaneous adipose tissue was obtained from three 5 ± 2 days calves, stored in phosphate-buffered saline (PBS) (HyClone, Logan, UT, USA) with 1% streptomycin and penicillin(Gibco, New York, NY, USA) and immediately brought back to the laboratory for isolation of primary adipocytes. In this experiment, bovine primary adipocytes were isolated using the tissue-block method. First, the fascia in the adipose tissue was removed with scissors and tweezers, and the adipose tissue block was carefully cut into small pieces of about 20 mm^3^ and placed in a new clean sterile 90 mm petri dish. The adipose tissue pieces were spread to the bottom of the petri dish at 1 cm intervals. The petri dish was placed upright for about 5 min to ensure that the adipose tissue did not fall off when it was inverted. The petri dish was carefully placed upside down at 37 °C, in a 5% CO_2_ incubator.

### 2.3. Prediction and Screening of miRNA

Basic information on the sequence and chromosomal location of circADAMTS16 was obtained via the NCBI online tool: “https://www.ncbi.nlm.nih.gov/ (accessed on 16 June 2021)”. miRNAs that bind to circADAMTS16 targets were predicted using RNAhybrid: “https://bibiserv.cebitec.uni-bielefeld.de/rnahybrid (accessed on 20 January 2022)” and miRanda: “http://www.microrna.org/microrna/home.do (accessed on 20 January 2022)” online software. VENNY 2.1 was used to intersect the online software: “https://bioinfogp.cnb.csic.es/tools/venny/ (accessed on 24 January 2022)”, and finally determine the potential target miRNA range of circADAMTS16.

### 2.4. Construction of Recombinant Plasmid and Luciferase Activity Assay

The full length of circADAMTS16 was amplified by using 2× Phanta Max Master Mix (Dye Plus) (Vazyme, Nanjing, China) and the amplified fragments were inserted into the pCD2.1 vector namely pCD2.1-circADAMTS16. The full-length circADAMTS16 sequence was inserted into a psiCHECK2 vector to obtain a dual fluorescent wild-type vector (psiCHECK-circADAMTS16-WT). Then, the mutated sequence at the miR-10167-3p binding site was inserted into psiCHECK2 to obtain a dual-fluorescent mutant vector (psiCHECK-circADAMTS16-Mut). HEK293T cells were spread into a 24-well plate, and when the cell density reached about 80%, the wild-type or mutant vector plasmid was transfected into HEK293T cells simultaneously with miR-10167-3p agomir. The relative activity of luciferase was calculated based on the normalization of firefly and renal luciferase activity using a Dual-Luciferase Reporter Assay System (Promega, Madison, WI, USA), with three parallel replicates per sample.

### 2.5. Cell Culture and Oil Red O Staining

Bovine primary adipocytes were isolated from bovine adipose tissues and cultured to 80% to 90% in growth medium (GM, DMEM + 10% fetal bovine serum) DMEM (HyClone, Logan, UT, USA), fetal bovine serum(Gibco, New York, NY, USA). Subsequently, the samples were dispersed into 6-well or 96-well plates for culture to a density of 80%, and the cells were treated and subjected to the corresponding assays. When the cell density reached more than 90%, the GM was replaced with induce medium (IM, GM containing 10 μg/mL of insulin, 1 μmol/L of dexamethasone, 0.5 mmol/L IBMX, and 1 μmol/L of rosiglitazone) for 2 days and then with maintenance medium (MM, GM containing 10 μg/mL of insulin and 1 μmol/L of rosiglitazone) every 2 days until cell maturation, insulin, dexamethasone, IBMX, rosiglitazone (Sigma-Aldrich, St. Louis, MO, USA). The culture medium in the petri dish of the cells which were to be stained was discarded, and the cells were washed three times with PBS, fixed with 10% formalin for 5 min, and incubated with the same volume of formalin at room temperature for 1 h. The cells were then washed three times with 60% isopropanol. After the petri dish was completely dried at room temperature, the cells were incubated with Red O working solution for 20 min at room temperature, and then washed three times with Oil Red O and PBS, followed by observation and photographing under an inverted microscope.

### 2.6. RNA Extraction, cDNA Synthesis, and qPCR

The total RNA in tissues and cells was extracted by the TriIzol method. The operating procedure was according to the instructions of the Trizol kit (Takara, Kyoto, Japan). The concentration (ng/μL) and OD 260/280 value of total RNA were detected by a multifunctional full-wavelength microplate reader. The integrity of total RNA was also detected by 1% agarose gel. Meanwhile, 1000 ng of the obtained total RNA was reversely transcribed into cDNA according to the random primers provided in the instruction of the reverse transcription kit and stored in a −20 °C refrigerator. qPCR was performed using 2× chamq universal SYBR qPCR master mix (Vazyme, Nanjing, China), using cell cDNA as the templates (three parallel replicates were set for each biological sample). The expression of miRNA was normalized to U6, and the expression of mRNA and circRNA were normalized to glyceraldehyde-3-phosphate dehydrogenase (GAPDH), primer information is shown in Table A1 in Appendix B.

### 2.7. EdU and CCK-8 Assay

Cell proliferation was detected using CCK-8 kits (Plomag, Beijing, China) and EdU kits (Ruibo, Guangzhou, China). In the CCK-8 assay, cells were placed in 96-well plates containing 100 μL of GM per well with six independent replicates per treatment group. Before detection, 10 μL CCK-8 reagent was added into each well, and the cells’ culture plate was gently shaken to make it uniform. After culturing in the incubator for 1 h, the absorbance at 450 nm was measured with a microplate reader (SYNERGY|LX, BioRad, Hercules, CA, USA). The cultured bovine primary adipocytes were spread into a 6-well cell-culture plate, and cell transfection was performed when the cell density reached about 80%. The transfection system was described in the lipo3000 (invitrogen, shanghai, China) instructions. The cell culture medium was replaced with OPTI-MEM(Gibco, New York, NY, USA) serum-free medium 30 min before transfection and solutions A and B were mixed and allowed to stand for 20 min. The mixed solutions were gradually added to the 6-well cell-culture plates, mixed gently, and put into a 37 °C, 5% CO_2_ cell incubator for 6 h, after which the growth medium with serum was replaced. After 48 h, the cells were collected for subsequent testing according to the EdU Test Kit instructions.

### 2.8. Flow Cytometry

#### 2.8.1. Cell Cycle Detection

Cell cycle and apoptosis were detected using a cell cycle kit and a cell apoptosis detection kit (Beyotime, Shanghai, China). The bovine preadipocytes cells were plated into 6-well plates and treated for 24 h. The cells were digested with trypsin, and centrifuged at about 1000× *g* for 3~5 min. Then, about 1 mL of pre-cooled PBS was added and the cells were resuspended. After re-centrifugation, the supernatant was discarded, and added into 1 mL of pre-cooled 70% ethanol, and gently blown and mixed. The cells were fixed at 4 °C for 24 h. After centrifugation at 1000× *g* for 3~5 min, the supernatant was discarded, and about 1 mL of pre-cooled PBS was added and the cells were resuspended. We added 0.5 mL of propidium iodide staining solution to each tube of cell samples, and the cell precipitate was slowly and fully re-suspended, incubated at 37 °C for 30 min in the dark, and detected by flow cytometry (BD-C6 Plus, Shanghai, China) (each treatment had three parallel replicates).

#### 2.8.2. Cell Apoptosis Detection

The cell culture medium was sucked out into a suitable centrifuge tube, the adherent cells were washed once with PBS, and a proper amount of trypsin cell digest solution (which may contain EDTA) was added to digest the cells. The digestion was stopped by adding the cell culture medium, which was blown down gently and transferred into a centrifuge tube. After the cells were centrifuged for 5 min at 1000× *g*, the supernatant was discarded and the cells were collected. The cells were then re-suspended gently with PBS and counted in a 50°C water bath. The cells were steamed for 2~3 min and subjected to apoptosis stimulation. Then, 50 to 100,000 cells were re-suspended and centrifuged at 1000× *g* for 5 min. The supernatant was discarded and the cells were resuspended gently with the addition of 195 µL AnnexinV–FITC conjugate and 5µl AnnexinV–FITC was added and mixed gently. Next, 10µL propidium iodide staining solution was added and mixed gently. After incubation in the dark at room temperature (20~25 °C) for 10–20 min, the apoptotic cells were detected (each treatment had three parallel replicates).

### 2.9. Data Analysis

For each group, at least three independent experiments were performed. The qPCR results of fluorescence quantification were analyzed by the 2^−ΔΔCt^ method, and the data are expressed as mean ± standard error (SEM). One-way analysis of variance (one-way ANOVA) using GraphPad Prism 8 software was carried out. *p* < 0.05 was considered statistically significant, indicated with an asterisk, while two asterisks indicate a *p* < 0.01.

## 3. Results

### 3.1. miR-10167-3p May Be a Target miRNA of circADAMTS16

In order to predict miRNAs, RNAhybrid, and miRanda were used online software. With the energy threshold less than −25.00 as the screening condition 18 miRNAs were screened out from RNAhybrid and 16 miRNAs were screened out from miRanda. The intersection of the two databases revealed seven potential target miRNAs: bta-miR-664a, bta-miR-2483-5p, bta-miR-365-5p, bta-miR-2407, bta-miR-2418, bta-miR-197, and bta-10167-3p (Figure 1a). MiRNAs with energy thresholds below -30.00 in both predictive software were selected for further validation (Figure 1b,c). circADAMTS16 overexpression significantly reduced miR-664a, miR-10167-3p, and miR-197 expression levels, while miR-365-5p expression was significantly increased (Figure 1b). Upon interference with circADAMTS16, miR-10167-3p expression was increased, while other miRNAs levels were significantly decreased (Figure 1c). These results suggested that miR-10167-3p may be a potential target miRNA of circADAMTS16. According to the prediction of bioinformatics analysis (Figure 1d), it is possible for a top-ranked complementary base in miR-10167-3p to act as a direct target of circADAMTS16 (Figure 1e). The constructed wild-type and mutant vectors were co-transfected into HEK293T cells with miR-10167-3p mimics, and the fluorescence values were detected by multifunctional full-wavelength microplate reader 48 h later. MiR-10167-3p luciferase activity was greatly hindered by wild-type circADAMTS16, rather than its mutant counterpart (Figure 1f).

### 3.2. circADAMTS16 Inhibits the Differentiation of Bovine Preadipocytes by Targeting miR-10167-3p

The role of circADAMTS16 in bovine adipocyte differentiation was investigated by performing gain-of-function experiments for bovine preadipocytes. After transfection of pCD2.1-circADAMTS16, the expression of circADAMTS16 was significantly up-regulated (Figure 2a), while the expression of adipogenic marker genes *PPARγ*, *C/EBPα,* and *C/EBPβ* was inhibited (Figure 2b). Then, miR-10167-3p agomir was transfected into bovine preadipocytes to increase the expression of *PPARγ*, lipoprotein lipase (*LPL*), and fatty-acid-binding-protein 4 (*FABP4*) (Figure 2c,d). Meanwhile, circADAMTS16 and miR-10167-3p agomir were co-transfected into bovine adipocytes and qPCR showed that circADAMTS16 reversed the promotion of miR-10167-3p agomir on adipogenesis (Figure 2c). The expression of circADAMTS16 inhibited the accumulation of lipid droplets in bovine adipocytes, while miR-10167-3p agomir promoted the accumulation of lipid droplets (Figure 2e). CircADAMTS16 may bind to miR-10167-3p as competitive endogenous RNA.

### 3.3. circADAMTS16 Promotes the Proliferation of Bovine Adipocytes by Targeting miR-10167-3p

We explored whether circADAMTS16 affects cell proliferation ability by targeting miR-10167-3p. Firstly, amplification of circADAMTS16 up-regulated the expression of two proliferation markers: cyclin-dependent-kinase 2 (*CDK2*) and cyclin-dependent-kinase 4 (*CDK4*) (Figure 3a,b). Furthermore, the opposite results were obtained in the loss-of-function experiment. The expression of *CDK2* and *CDK4* was down-regulated by interfering with circADAMTS16, while the expression of *CDK2*, *CDK4*, proliferating cell nuclear antigen (*PCNA*), and cyclin D2 (*CCND2*) was up-regulated by interfering with miR-10167-3p (Appendix A). Following co-transfection of circADAMTS16 and miR-10167-3p agomir, it was found that circADAMTS16 alleviated the inhibitory effect of miR-10167-3p agomir on cell proliferation (Figure 3c). Cells transfected with pCD2.1-circADAMTS16 showed significantly higher cell viability at 12–48 h than control cells, while the cell viability was inhibited after transfection with miR-10167-3p agomir (Figure 3d,e). Similarly, cell viability was inhibited after transfection of si-circADAMTS16, whereas it was enhanced after transfection of miR-10167-3p antagomir (Appendix A). Meanwhile, EdU experiments further verified that circADAMTS16 targeted miR-10167-3p to promote the proliferation of bovine adipocytes (Figure 3f,g). Furthermore, cell cycle analysis showed that circADAMTS16 promoted the transformation of bovine adipocytes from G0/G1 phase to S phase (Figure 3h), while miR-10167-3p inhibited the transformation of bovine adipocytes from G0/G1 phase to S phase (Figure 3i). Loss-of -function experiments also yielded the opposite results. After transfection of miR-10167-3p antagomir, EdU, and flow cytometry detection of cell proliferation showed that inhibition of miR-10167-3p promoted adipocyte proliferation (Appendix A). Taken together, the above results indicate that circADAMTS16 can promote adipocyte proliferation by targeting miR-10167-3p.

### 3.4. circADAMTS16 Inhibits Bovine Adipocytes Apoptosis by Targeting miR-10167-3p

Similarly, we also explored whether circADAMTS16 can affect apoptosis by targeting miR-10167-3p. Overexpression of circADAMTS16 inhibited *Caspase-9-* and *Bcl-2-*associated X protein (*Bax*) expression and promoted the expression of *Bcl-2*, whereas transfection of miR-10167-3p agomir into bovine adipocytes up-regulated the expression of *Caspase-3*, *Caspase-9,* and *Bax* (Figure 4a,b). After interference with circADAMTS16 and miR-10167-3p, their expression levels were opposite (Appendix A). However, when circADAMTS16 and miR-10167-3p agomir were co-transfected into bovine adipocytes, it was found that circADAMTS16 alleviated the up-regulated expression of *Caspase-9* and *Bax* by miR-10167-3p (Figure 4c). The flow cytometry results indicated that interference with circADAMTS16 and forced expression of miR-10167-3p both promoted apoptosis (Figure 4d), overexpression of circADAMTS16, and interference with miR-10167-3p inhibited apoptosis (Appendix A). However, apoptosis was effectively alleviated by forced expression of circADAMTS16 and miR-10167-3p (Figure 4d–f).

## 4. Discussion

There are two types of adipose tissue growth: hyperplasia (increasing the number of adipocytes) and hypertrophy (increasing the size of adipocytes).These are important endocrine organs for metabolism [31]. Fat content is one of the factors that affect the quality of meat. Intramuscular fat affects the tenderness, juiciness, and flavor of meat as well as the sensory evaluation of consumers. An appropriate increase in intramuscular fat content will result in an increase in muscle tenderness, a decrease in shear force, a brighter muscle, a lighter color, and an increase in marbling [32]. With the deepening of research on ncRNA, it has been found that it plays an important role in the regulation of lipid metabolism. NcRNA is a type of functional RNA molecule that is not translated into protein [33]. According to high-throughput sequencing data of circRNAs in bovine adipose tissue, circADAMTS16 is differentially expressed. However, circADAMTS16 has not been reported in fat development. Therefore, we explored the role of circADAMTS16 in the differentiation and proliferation of bovine adipocytes.

Studies have shown that circRNAs can regulate mRNA expression level by targeting miRNA [34]. We validated the targeting relationship of circADAMTS16 to miR-10167-3p. We found that overexpression of circADAMTS16 not only reduced lipid droplet accumulation phenotypically, but also suppressed *PPARγ*, *C/EBPα,* and *C/EBPβ* expression at the transcriptional level. In avian adipogenesis, *C/EBPα* accumulates genes related to lipolysis, lipogenesis, and fatty-acid desaturation, while C/EBPβ, δ accumulates genes related to de novo lipogenesis and fatty-acid elongation [35]. The results showed that the protein arginine methyl transferase 7 (PRMT7) interacted with C/EBPβ and methylated during adipogenesis, and regulated the accumulation of C/EBPβ in its PPAR-γ2 promoter target [36]. However, FoxO1- and C/EBPβ-binding may regulate pre-lipogenesis through the *C/EBPβ-FoxO1-C/EBPβ* feedback regulatory loop and the FoxO1-C/EBPβ protein complex, and *FoxO1* and *C/EBPβ* co-knockout promote lipogenesis in adipocytes [37]. C/EBPα and FoxO1 ingest miR-144, forming a protein complex that binds to *AdipoQ* promoter, thus regulating *AdipoQ* transcription to inhibit fat formation [38]. It is well known that *PPARγ* can activate that expression of adipocyte differentiation regulators such as *FABP4*, *C/EBPs,* and *LPL* to promote adipocyte differentiation for fat deposition [39]. *LPL* is one of the early markers of preadipocyte differentiation into adipocytes, and reaches a stable level in mature adipocytes [40]. The binding of plasma triglyceride fatty acids to white adipose tissue is LPL-dependent and the enzyme is regulated by angiopoietin-like protein-4 (ANGPTL-4), an unfolding molecular chaperone that converts active LPL dimers to inactive monomers [41]. Feng et al. [24] reported that circMARK3 promoted the adipogenic differentiation of buffalo adipocytes and 3T3-L1 cells by up-regulating the expression levels of lipogenic marker genes *PPARγ*, *C/EBPα*, and *FABP4*. *FABP4* is involved in fatty acid transport and plays an important role in animal fat deposition [42]. These results suggest that circADAMTS16 inhibits bovine preadipocyte differentiation by targeting miR-10167-3p.

Previous studies have shown that *ADAMTS16* expression inhibits the proliferation of tumor cells [43], whereas our data suggested that circADAMTS16 promotes adipocyte proliferation. Similarly, Guarnerio J et al. [44] reported that the *Zbtb7a* gene encodes circPOK, yet circPOK is contrary to the mRNA function of the linear *Zbtb7a* gene. Overexpression of circADAMTS16 significantly increased the expression of proliferation marker genes *CDK2* and *CDK4*, while overexpression of miR-10167-3p inhibited their expression. The CDKs family participated in the regulation of basic cellular activities, including proliferation and transcription, and controlled the transformation between different stages of the cell cycle [45]. A variety of CDKs inhibitors have been used for the treatment of cancer diseases [46]. Flow cytometry analysis showed that circADAMTS16 promoted the transition from G0/G1 phase cells to S phase cells. It is thus suggested that circADAMTS16 promotes adipocyte proliferation through targeted binding to miR-10167-3p. There are two main stages in the cell cycle: interphase and mitosis [47]. Cells spend most of their life in the interphase, which is divided into three stages: gap 1 (G1), synthesis (S), and gap 2 (G2). In the interphase, the cells are growing and ready to divide, the fourth stage, gap zero (G0), describes resting cells and cells that rarely or never divide [48]. Studies have shown that *C/EBPα* promotes the proliferation of adipocytes by mediating the MSTRG.12568.2/FoxO3 trans-activation of serine/threonine/tyrosine interacting protein gene (*STYX*) [49]. This may be the reason for the down-regulation of *C/EBPα* expression after transfection of miR-10167-3p. The research by Chang et al. [50] indicated that lipopolysaccharide promoted the proliferation of preadipocytes and lipogenesis through Janus kinase (JAK)/signal transducer and activator of transcription (*STAT*) and AMPK-regulated cytoplasmic phospholipase *(CPLA2*).

Apoptosis is critical for mammalian tissue homeostasis and preadipocytes can also undergo apoptotic cell death [51]. Mature adipocytes are less sensitive to apoptotic stimuli than preadipocytes. The increase in the number of adipocytes is achieved through the replication and differentiation of preadipocytes, and the decrease in the number of adipocytes is achieved through the apoptosis of preadipocytes and adipocytes and possible dedifferentiation of adipocytes [52]. Cell proliferation and apoptosis are the basis of biological growth and reproduction, and the most basic process in the biological system [53]. Many physiological processes, including proper tissue development and in vivo balance, require a balance between apoptosis and cell proliferation [54]. In the early stage of apoptosis, the level of intracellular reactive oxygen species (ROS) increases [55]. *Caspase-9* is necessary for mitochondrial morphological changes and ROS production. It cleaves the Bid and activates it as tBid. After being activated by *Caspase-9*, *Caspase-3* inhibits the production of ROS, which is necessary for effective implementation of apoptosis [56]. Mitochondria play a decisive role in the regulation of apoptosis and necrotic cell death [57]. Overexpression of *Bcl-2* in pre-adipocytes decreases its susceptibility to apoptosis at the level of mature adipocytes. During lipogenesis, the balance of pro-apoptotic and anti-apoptotic molecules is altered, resulting in higher resistance to apoptosis [58]. Among the involved mechanisms, the transformation of mitochondrial permeability seems to be mainly related to necrosis, while the release of caspase-activating protein in the early stage of apoptosis is mainly regulated by the Bcl-2 protein family [59]. Overexpression of *Bcl-2* inhibits apoptosis by interacting with pro-apoptotic genes [60]. In addition, the overexpression of *BAX* overcomes the effects of anti-apoptotic proteins, and eventually leads to apoptosis [61]. It is speculated that miR-10167-3p promotes the expression of *Bcl-2* and promotes cell apoptosis.

## 5. Conclusions

In conclusion, we confirmed that circADAMTS16 inhibits bovine adipocyte differentiation and promotes bovine adipocyte proliferation by targeting miR-10167-3p. Furthermore, circADAMTS16 inhibits apoptosis by targeting miR-10167-3p. Therefore, circADAMTS16 may become another key factor in the regulation of fat deposition in bovine.

## Figures and Tables

**Figure 1 cells-12-01175-f001:**
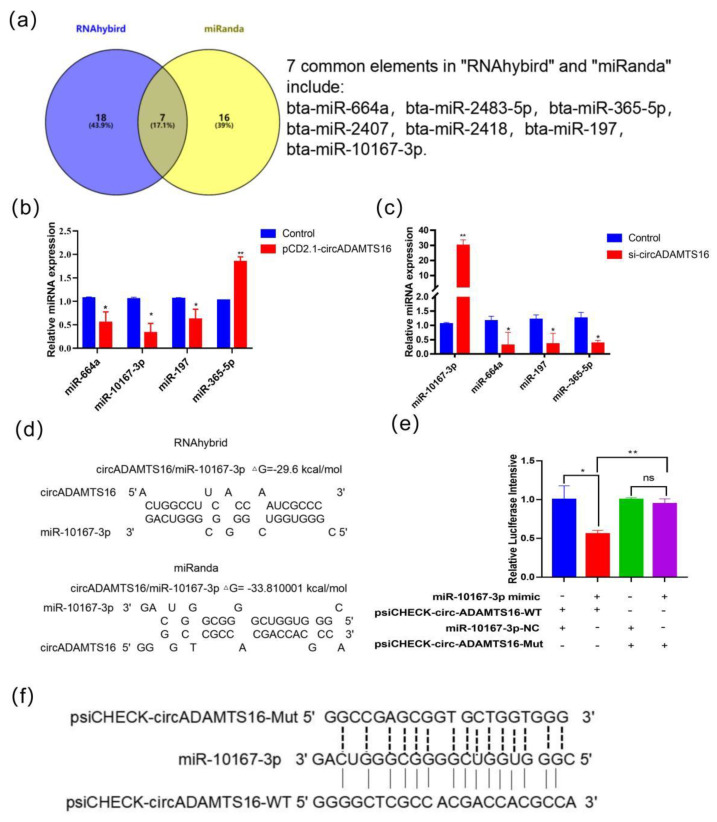
Identification of potential target miRNA of circADAMTS16. (**a**) RNAhybrid and miRanda predict potential miRNAs. (**b**,**c**) The expression of potential miRNAs was detected by qPCR. (**d**) RNAhybrid and miRanda online software were used to predict the targeted binding site of miR-10167-3p and circADAMTS16. (**e**) circADAMTS16 and miR-10167-3p wild-type vector binding site and mutant type vector mutation site. (**f**) Dual luciferase reporting experiment verified the targeting relationship between miR-10167-3p and circADAMTS16. * *p* < 0.05, ** *p* < 0.01, ns indicated *p* > 0.05.

**Figure 2 cells-12-01175-f002:**
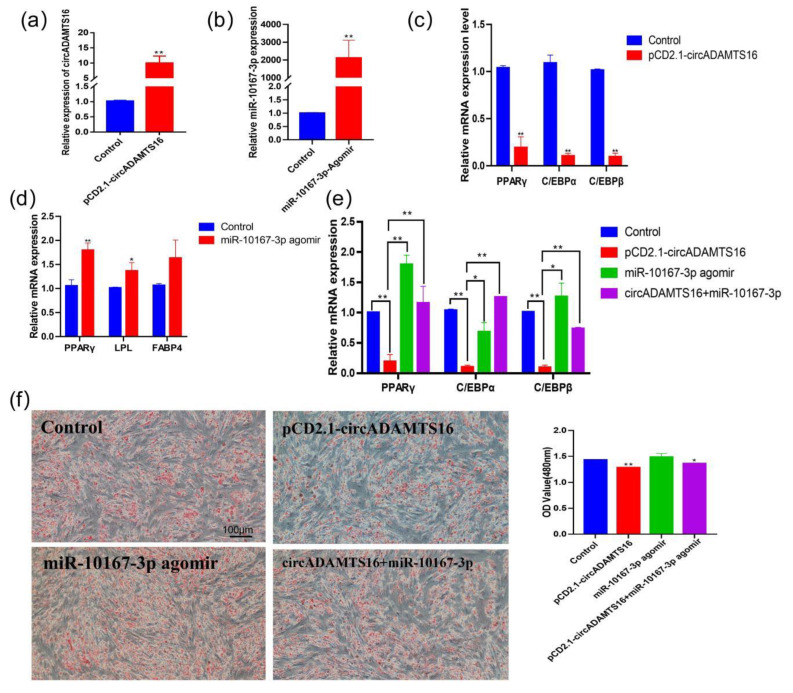
circADAMTS16 targeting miR-10167-3p inhibits differentiation of bovine preadipocytes. (**a**) Expression of circADAMTS16 after transfection of pCD2.1-circADAMTS16. (**b**) Expression of miR-10167-3p after transfection of miR-10167-3p agomir. (**c**–**e**) Expression levels of *PPARγ*, *C/EBPα*, *C/EBPβ*, *LPL*, and *FABP4* after transfection of circADAMTS16, miR-10167-3p agomir, and circADAMTS16 + miR-10167-3p. (**f**) The negative control, pCD2.1-circADAMTS16, miR-10167-3p agomir, and pCD2.1-circADAMTS16 + miR-10167-3p agomir were transfected into bovine preadipocytes. After eight days of induced differentiation, Oil Red O staining was performed, and the histogram of Oil Red O staining was quantitatively drawn by spectrophotometry. Scale bar indicates 100 µm * *p* < 0.05, ** *p* < 0.01.

**Figure 3 cells-12-01175-f003:**
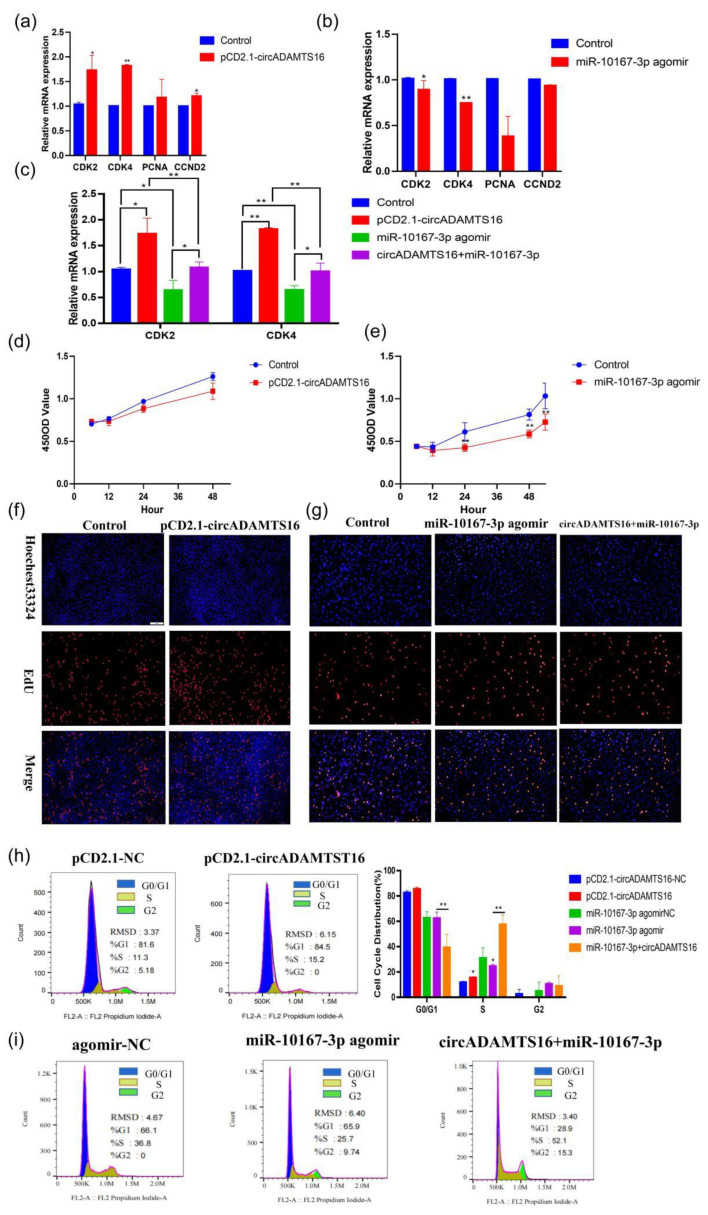
circADAMTS16 promotes the proliferation of bovine adipocytes by targeting miR-10167-3p. (**a**,**b**) qPCR was used to detect the mRNA expression levels of proliferation genes *CDK2* and *CDK4* after transfection of pCD2.1-circADAMTS16 and miR-10167-3p agomir, respectively. (**c**) the expression levels of proliferation genes *CDK2* and *CDK4* were detected after co-transfection of circADAMTS16 and miR-10167-3p agomir into bovine adipocytes. (**d**,**e**) The viability of pCD2.1-circADAMTS16 and miR-10167-3p agomir transfected cells was detected by CCK-8. (**f**,**g**) Cell proliferation was detected by EdU. (**h**,**i**) Cell cycle was analyzed by flow cytometry (pCD2.1-NC was empty, and agomir-NC was used as the control of specially synthesized agomir). Scale bar indicates 200 µm. * *p* < 0.05, ** *p* < 0.01.

**Figure 4 cells-12-01175-f004:**
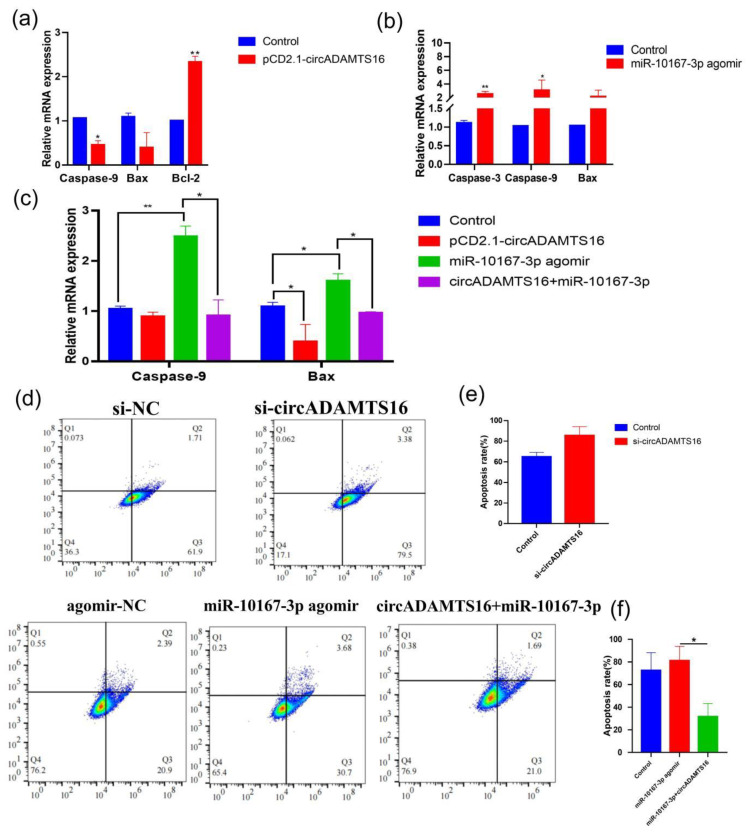
circADAMTS16 inhibits bovine adipocytes apoptosis by targeting miR-10167-3p. (**a**,**b**) The mRNA expression levels of apoptotic genes were detected after transfection of pCD2.1-circADAMTS16 and miR-10167-3p agomir into bovine adipocytes. (**c**) The expression levels of *Caspase-9* and *Bax* were detected after co-transfection of circADAMTS16 and miR-10167-3p agomir into bovine adipocytes. (**d**) Apoptosis was detected by flow cytometry (si-NC was the synthetic negative control). (**e**,**f**) Apoptosis rate. * *p* < 0.05, ** *p* < 0.01.

## Data Availability

The data presented in this study are available in the article.

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
