# Peer review of "CircADAMTS16 Inhibits Differentiation and Promotes Proliferation of Bovine Adipocytes by Targeting miR-10167-3p"

_cells, 2023, doi:10.3390/cells12081175_

Round 1

Reviewer 1 Report (New Reviewer)

Manuscript needs re-reading for spell check, duplications, phrasing errors. 

Author Response

Point1. Manuscript needs re-reading for spell check, duplications, phrasing errors.

Response1:Thank you very much for your valuable comments, which has greatly helped to improve our manuscript.

In response to your suggestions, we re-read the manuscript and check for and correct misspellings, repetitions, and grammatical errors that appear in the manuscript.

Finally, thank you once again for your time and energy for our article. Due to your valuable comments, we have the opportunity to improve the manuscript, and the addition of content has laid the foundation of the work carried out in this manuscript, which makes the structure of this manuscript more complete.

Reviewer 2 Report (New Reviewer)

The study aims to clarify the functions of circADAMTS16 on bovine preadipocytes, which the authors found in the previous study. The prediction and analysis were performed to screen the target miRNA, and the effects of circADAMTS16 and miR-10167-3p on cell proliferation and apoptosis through multiple experiments were validated. The above results provide evidence to elucidate the inhibitory effect of circRNA on bovine adipocyte differentiation.

1. Lines 114-115 Please provide a brief procedure for isolating the cells or a reference to the literature. In addition, the authors state that multiple markers were used to validate the cells; 2. Please provide this in the supplementary material.

3. Please list these seven miRNAs in the results in line 220.

4. Figure 4 needs a clear layout and design for readability.

5. The result description does not match FigS2-e; please double-check the legend for figS2-E.

6. Please describe or define the control in detail in the figures like figure 3h.

7. Primer information is incomplete; please add all primers sequence, including PPARγ, circADAMTS16, and miR-10167-3p.

Author Response

Response to Reviewer 2 Comments

Point 1-2:Lines 114-115 Please provide a brief procedure for isolating the cells or a reference to the literature. In addition, the authors state that multiple markers were used to validate the cells; 2. Please provide this in the supplementary material.

Response 1-2:Thank you very much for your valuable comments, which has greatly helped to improve our manuscript.

The adipocyte isolation step has been supplemented in the manuscript at lines 118-130. The supplementary content is as follows: In this experiment, bovine primary adipocytes were isolated using the tissue block method. First, the fascia in the adipose tissue was removed with scissors and tweezers, and the adipose tissue block was carefully cut into small pieces of about 20 mm3 in a clean sterile petri dish. A new 90 mm petri dish was taken. The cut adipose tissue was spread to the bottom of the petri dish at an interval of 1 cm. The petri dish was placed upright for about 5 min to ensure that the adipose tissue did not fall off when it was inverted. The petri dish was carefully placed upside down at 37 ℃, In a 5% CO2 incubator, the petri dish was placed immediately after 6 h and 10 mL of high-glucose medium containing the mixture of 10% fetal bovine serum and 1% streptomycin was added. Observing whether adipocytes are free after 48 h of culture; When more adipocytes were released around most of the tissue blocks, the tissue blocks were gently picked out with a 1 mL spear head, and the adherent cells were digested with 0.25% trypsin and then re-inoculated into new petri dishes for culture.

In addition, isolated adipocytes were not validated with multiple markers in this assay. The fat cells of the three separated cattle were labeled and stored in liquid nitrogen. I am sorry for the misunderstanding we have brought to you due to the lack of clarity in our statement. We've made changes in the manuscript, at lines 130-132. Revised as:The separated three bovine adipocytes were marked and stored in liquid nitrogen for subsequent experiments.

Point 3:Please list these seven miRNAs in the results in line 220.

Response 3: Thank you for your valuable comments on improving our manuscript. Seven miRNA have been listed in line 240-242 of the manuscript based on your comments.

Point 4: Figure 4 needs a clear layout and design for readability

Response 4: Thank you for your valuable comments. For your comments, we adjusted the layout and design of figure 4, please refer to it!

Point 5: The result description does not match figS2-e; please double-check the legend for figS2-E.

Response 5: Thank you very much for your comments and we are sorry to have brought you into question due to our negligence. We re-checked the supplementary figure 2, modified the wrong picture and re-adjusted the layout, and made changes to the legend, please refer to!

Point 6:Please describe or define the control in detail in the figures like figure 3h.

Response 6:Thank you for your valuable comments. In response to your comments, we modified the manuscript on lines 320-321 and 342, with a detailed description of the controls in the legend.

Point 7:Primer information is incomplete; please add all primers sequence, including PPARγ, circADAMTS16, and miR-10167-3p.

Response 7:Thank you for your reminder.We have related primer information to supplement, please refer to!

Finally, thank you once again for your time and energy for our article. Due to your valuable comments, we have the opportunity to improve the manuscript, and the addition of content has laid the foundation of the work carried out in this manuscript, which makes the structure of this manuscript more complete.

Reviewer 3 Report (New Reviewer)

Comments and Suggestions for Authors

In the manuscript “CircADAMTS16 inhibits differentiation and promotes prolifer- 2

ation of bovine adipocytes by targeting miR-10167-3p”authors described the function and possible mechanisms of a circular RNA circADAMTS16 in bovine fat deposition. Circular RNAs have been observed in many species and tissues and are now recognized as an explicit component of the transcriptome, especially with developments in high-throughput RNA-seq that have revealed a significant amount of circular RNAs. However, their functions are still murky. This study identified the high expression levels of circADAMTS16 in bovine adipose tissue and revealed that circADAMTS16 regulate the proliferation of bovine adipocytes by targeting miR-10167-3p.

The topic may be of interest. however, several inaccuracies may be found throughout the paper. Please find below specific points:

1.     Figure 1d. The binding site of circADAMTS16 and miR-10167-3p predicted by RNAhybird and miRanda seems to be inconsistent. How to explain this? At the same time, RNAhybird predicted miR-10167-3p, while miRanda predicted miR-10169-3p. Why?

2.     I consider that the significance marking in Figure 2e and Figure 3c should refer to the marking method in Figure 1e.

3.     Line 246 “PPAR γ” or Line 248 “PPARγ” should be unified.4. Figure 2 There is no communication between drawing notes and drawings.

4. Figure 2 There is no communication between drawing notes and drawings.

5. Figure 2f. Scale?

6. Figure 3b CDK2 and Figure 3e 48h, adjust the "*" position (upward).

Author Response

Response to Reviewer 3 Comments

Point1:Figure 1d. The binding site of circADAMTS16 and miR-10167-3p predicted by RNAhybird and miRanda seems to be inconsistent. How to explain this? At the same time, RNAhybird predicted miR-10167-3p, while miRanda predicted miR-10169-3p. Why?

Response1: Thank you very much for your valuable comments, which has greatly helped to improve our manuscript.

Many circRNA databases that can predict the combined miRNAs are operated based on various known databases of related miRNAs, and the principles of different program predictions are different, so the two prediction softwares predicted that the binding sites of miR-10167-3p and circADAMTS16 were inconsistent. In addition, you mentioned that "miRanda predicts miR-10169-3p", which is a typographical error, and we are sorry to have caused you to have doubts due to our negligence. We have made changes in the picture.

Point2: I consider that the significance marking in Figure 2e and Figure 3c should refer to the marking method in Figure 1e.

Response2: Thank you for your valuable comments on improving our manuscript. We have further improved figure 2e and figure 3c according to your opinions. Please refer to them!

Point3-6: Line 246 “PPARγ” or Line 248 “PPARγ” should be unified. Figure 2 There is no communication between drawing notes and drawings. Figure 2f. Scale?

Response3-6:Thank you for pointing out the problems in our manuscript, which will help us to improve the manuscript.

In response to your question that there is no unification of PPARγ in lines 246-248, we have made a modification, located in line 282 of the manuscript. In addition, the annotated portion of figure 2 has been modified based on the picture content and is located on lines 280-288 of the manuscript. At the same time, the scale of figure 2f is supplemented. The "*" in figure 3b and figure 3e has been moved up based on your opinion. Please consult!

Finally, thank you once again for your time and energy for our article. Due to your valuable comments, we have the opportunity to improve the manuscript, and the addition of content has laid the foundation of the work carried out in this manuscript, which makes the structure of this manuscript more complete.

This manuscript is a resubmission of an earlier submission. The following is a list of the peer review reports and author responses from that submission.

Round 1

Reviewer 1 Report

Adipogenesis plays a vital role in the bovine fat deposition, and many circRNAs have emerged as important regulators in various tissues and cell types. Hu et al. examined the regulatory mechanism of circADAMTS16 and miR-10167-3p in bovine adipocytes. This study in overall is comprehensive and significant, and certainly also fit the highlighted scope of the journal. However, several points need to be improved.

1. Line 75: Is there a reference for the specific expression of circADAMTS16 in bovine adipose tissue? If not, could the author validate the expression of circADAMTS16 using experimental techniques such as qPCR?

2. Figures 1 and 2 are in the wrong order.

3. Is the qPCR data in Figure 2b, 2d, and 2e duplicated? The duplicate data could be merged.

4. Line 190, Figure 2e: Mir-10167-3p did not increase the expression of CEBPα. Even, the sample co-transfected with circADAMTS16 and miR-10167-3p agomir displayed a higher expression of CEBPα than the sample transfected with miR-10167-3p agomir. The author needs to discuss more about the inconsistent results of CEBPα.

5. What is the association between adipogenesis and apoptosis in bovine adipocytes? Could the author perform more experiment to explore this?

Author Response

Please check the responses attached.

Reviewer 2 Report

This paper deals with the possible role of CircADAMTS16 in the process of bovine lipid metabolism. The results found are potentially interesting, but some concerns should be addressed before it is accepted for publication.

1-      The authors used the DDCt method to perform real-time PCR. A premise to using this method is that both target and reference (internal control) genes must have similar efficiency (E) of amplification, calculated based on the slope of the calibration curves of each gene. Thus, does the E of the qPCR was equivalent for the genes analyzed? Please, provide this information.

2-      Authors should provide information regarding this notion. I also wonder how authors evaluate the appropriateness of GAPDH as a suitable reference gene.

3-      P53 is crucial in the regulation of apoptosis and its signaling is important during In vitro adipogenic differentiation. Why did not the authors evaluate P53 in the current study?

4-      The authors attempt to evaluate apoptosis by assessing some apoptotic markers, such as Bcl2, Bax, caspase 3, and caspase 9. However, there is only mRNA expression data for these markers. It is recommended to include protein analysis, such as Western Blot, for some markers.

5-      There is some opposing evidence on the role of PPARγ and CCAAT enhancer binding protein α and beta in the context of adipogenesis. It is highly recommended that authors point to available controversial evidence when the results were discussed.

Author Response

Response to Reviewer 2 Comments

We thank the reviewers for their suggestions and comments. We hereby reply to the comments made by the reviewers as follows:

Reviewers1

Point1: The authors used theDDCt method to perform real-time PCR. A premise to using this method is that both target and reference (internal control) genes must have similar efficiency (E) of amplification, calculated based on the slope of the calibration curves of each gene. Thus, does the E of the qPCR was equivalent for the genes analyzed? Please, provide this information.

Response1:Sorry for the confusion brought to the reviewers. The amplification sample is diluted 10 times from the original concentration, and the number of amplification cycles is below 40 cycles. The amplification curve is shown in the figure. 2n=10,n=3.32, E=10-1/slope, 2=10-1 slope, slope =-3.32, efficiency =(E-1)×100%=(2-1)×100%=100%. See cover letter for picture

Point2:Authors should provide information regarding this notion. I also wonder how authors evaluate the appropriateness of GAPDH as a suitable reference gene.

Response2:Thank you very much for your question. GAPDH is a housekeeping gene that is expressed at a high level in almost all tissues, and its expression in the same kind of cells or tissues is generally constant. After many experiments conducted by our team, we found that GAPDH was the most stable internal reference gene, so we selected it as the internal reference gene. Example:Lei, Z., Wei, D., Ma, Y., Tang, L., Wang, S., Wang, P., et al., 2022. miR-302b promotes bovine preadipocyte differentiation and inhibits proliferation by targeting CDK2. Anim Biotechnol:1-8.

Point3: P53 is crucial in the regulation of apoptosis and its signaling is important during In vitro adipogenic differentiation. Why did not the authors evaluate P53 in the current study?

Response3:Thank you very much for your suggestion. Studies have shown that DNA repair is activated early in p53-induced apoptosis, i.e., p53-induced apoptosis is reversible. P53 activation induces phosphatidylserine (PS) externalization early in apoptosis, and these early apoptotic cells with externalized PS can be saved and proliferated if the apoptotic stimulus is removed. In addition, p53 partially inhibits preadipocyte differentiation and adipogenesis by regulating adipocyte gene expression and akt signaling pathway. Therefore, this study did not evaluate the expression of p53 when evaluating apoptosis.

Point4: The authors attempt to evaluate apoptosis by assessing some apoptotic markers, such as Bcl2, Bax, caspase 3, and caspase 9. However, there is only mRNA expression data for these markers. It is recommended to include protein analysis, such as western blot, for some markers.

Response4:Thank you very much for your suggestion. In this study, we analyzed the effect of circADAMTST6 targeting miR-10167-3p on the expression level of apoptotic genes, and further verified the apoptosis by flow cytometry.

Point5: There is some opposing evidence on the role of PPARγ and CCAAT enhancer binding protein α and beta in the context of adipogenesis. It is highly recommended that authors point to available controversial evidence when the results were discussed.

Response5:Thank you very much for your suggestion. PPARγ, C/EBPα, and C/EBPβ are crucial during adipocyte differentiation. A small number of studies have shown that they also have different roles in adipocyte differentiation, which were added in the discussion according to the opinions of reviewers. See lines 285-290 in the discussion section of the specific manuscript.(in yellow)

Round 2

Reviewer 1 Report

1.  There is not a reference or exprimental results for the specific expression of circADAMTS16 in bovine adipose tissue.

2. Line 75: "Sequencing showed that circADAMTS16 was specifically pressed in bovine adipose tissue" What is the Sequencing result?

3. The author do not provide any expriements about the association between adipogenesis and apoptosis in bovine adipocytes. The results of apoptosis part seems not nessary for the conclusion of this study "CircADAMTS16 inhibits differentiation and promotes proliferation of bovine adipocytes by targeting miR-10167-3p".

Author Response

Please check the responses attached.

Reviewer 2 Report

All comments have been responded and the current version of the manuscript is acceptable.

Author Response

Thank you very much for your suggestions and opinions in your busy schedule. Best wishes.

Round 3

Reviewer 1 Report

1. The tissue expression profile of circADAMTS16 provided in response 1 showed that circADAMTS16 was not specifically expressed in adipose tissue.

2. In the reponse, the author mentioned that the sequencing data is being uploaded. However, no sequencing result was avaliable in the manuscript.

  •